# Conformational Transition of Semiflexible Ring Polyelectrolyte in Tetravalent Salt Solutions: A Simple Numerical Modeling without the Effect of Twisting

**DOI:** 10.3390/ijms25158268

**Published:** 2024-07-29

**Authors:** Dan Lu, Aihua Chai, Xiuxia Hu, Peihua Zhong, Nianqian Kang, Xianfei Kuang, Zhiyong Yang

**Affiliations:** 1Department of Physics, Jiangxi Agricultural University, Nanchang 330045, China; pitlu@126.com (D.L.); huxiuxia_21@163.com (X.H.); mico19992000@163.com (N.K.); kuangxianfei@126.com (X.K.); 2College of Data Sience, Jiaxing University, Jiaxing 314001, China; ahchai@mail.zjxu.edu.cn; 3College of Computer Information and Engineering, Jiangxi Agricultural University, Nanchang 330045, China; phzhong@163.com

**Keywords:** semiflexible ring polyelectrolyte, tetravalent salt solution, molecular dynamics, Debye length, phase transition

## Abstract

In this work, the conformational behaviors of ring polyelectrolyte in tetravalent salt solutions are discussed in detail through molecular dynamics simulation. For simplification, here we have neglected the effect of the twisting interaction, although it has been well known that both bending and twisting interactions play a deterministic in the steric conformation of a semiflexible ring polymer. The salt concentration *C_S_* and the bending energy *b* take a decisive role in the conformation of the ring polyelectrolyte (PE). Throughout our calculations, the *b* varies from *b* = 0 (freely joint chain) to *b* = 120. The salt concentration *C_S_* changes in the range of 3.56 × 10^−4^ M ≤ *C_S_* ≤ 2.49 × 10^−1^ M. Upon the addition of salt, ring PE contracts at first, subsequently re-expands. More abundant conformations are observed for a semiflexible ring PE. For *b* = 10, the conformation of semiflexible ring PE shifts from the loop to two-racquet-head spindle, then it condenses into toroid, finally arranges into coil with the increase of *C_S_*. As *b* increases further, four phase transitions are observed. The latter two phase transitions are different. The semiflexible ring PE experiences transformation from toroid to two racquet head spindle, finally to loop in the latter two phase transitions. Its conformation is determined by the competition among the bending energy, cation-bridge, and entropy. Combined, our findings indicate that the conformations of semiflexible ring PE can be controlled by changing the salt concentration and chain stiffness.

## 1. Introduction

Polyelectrolytes (PEs) such as proteins [1] and DNA [2] are ubiquitous in nature. Besides being essential for human physiology and metabolism, they play a fundamental role in everyday life and are used in a very wide range of technical applications such as drug and gene delivery, tissue engineering, and battery electrolytes. The description of polyelectrolytes is perhaps the most challenging subject today among all chemical and biological systems in their liquid and solid states. The main reason is that there are too many variables that control the polyelectrolyte phenomena. The conformation of PE chains determines the functionality of PE systems. One can control the conformation of PE and adjust the self-assembly process by altering the factors such as the valency of counterions and charge fractions, the temperature [3], Ph [4], salt concentration [5], etc., which in turn change biofunctional processes. Salts are also ubiquitous in nature, and there are multiple types of salts. They produce profound effects on biological processes on the molecular scale, from molecular interactions, to mechanical properties, to folding, to self-organization and assembly, to energy and material transport etc. Polyelectrolytes in electrolyte solutions are fundamentally important in biology [6]. The behavior of PEs is greatly affected by ions in the solutions, because of ionic redistribution and the screening effect. The interaction between salt ions and polyelectrolytes is a long-studied field which is full of important open challenges in polymer science and technology [7,8].

The ionic strength of the solution plays a very important role in both molecular and cellular levels in biological systems. For instance, inorganic salts change tissue permeability. Salt ions influence the stability of their conformations and solubility of proteins. The fast and reversible collapse of the DNA molecule which responds to the salt-induced condensation process provides an efficient way of gene delivery in genetic therapy. Interactions between counterions and ionized PE chains play an important role in water ultrafiltration through PE membranes. When the system of surfactants and PEs form micelle-like structures, the coagulation efficiency of PEs for water purification can be improved at low salt concentrations and critical aggregation concentration lower, while screening leads to a larger critical aggregation concentration at higher concentrations. Therefore, it is very important for many applications to understand how the behavior of PEs is modified in salt solutions.

The monovalent cations screen intrachain electrostatic repulsions, and cause the PE chain to coil and reach its unperturbed dimension [9]. Comparing multivalent ions with monovalent ions, the effects of the multivalent ions are frequently not only stronger (due to the obviously higher charge), but also qualitatively different. According to the existing theories [10,11,12,13] and simulations [14,15,16], multivalent cations show a stronger tendency to form a PE complex, also, a stronger bridging of charged groups is formed, compared with monovalent ions, as fewer ions can drive them form a more stable and favorable correlated system with the charged chains. Condensing DNA by multivalent counterions such as Cohex+3, Spd+3 or Spm+4 becomes one of the most interesting and important electrostatic aspects of the DNA system. The non-monotonic effect of multivalent counterions on DNA ejection from viruses is expected to have the same physical origin as the phenomenon of reentrant DNA condensation in free solution in the presence of counterions of tri-, tetra-, and higher valence. In addition, the multivalent ions are used to condense DNA in nature for packaging inside cells and viruses [17,18]. Thus, scientists pay more attention to the study of PEs in solutions of multivalent salt.

Polymeric systems with charges show complex conformational and dynamic behavior due to the interplay between multivalent ions and PEs. There have been many experiments [9,19,20,21,22,23,24], simulations [25,26,27], and theories [28,29] focusing on studying the behavior of PEs in multivalent salt solutions. Minko et al. found that the collapse of PE coils is steeper for multivalent counterions, compared with monovalent counterions, however, the re-expansion is only incremental in the presence of multivalent counterions, using in situ atomic force microscopy [19]. Rydzek et al. found that PMAA chains aggregate together when the cation exchange from Na^+^ to Cu^2+^ [23]. Based on a coarse-grained model of a nucleic acid, Hsiao et al. studied the effect of ions of different sizes and charge on its conformational behavior. In the presence of tetravalent ions, the conformation of PE is extended at low and high salt concentration while they fold into a condensed structure at an intermediate salt concentration. They found that the behavior can be regulated by the overcharging of the nucleic acid at high salt concentrations [30,31]. Wang et al. found that it is more effective for multivalent ions to induce ssDNA strands collapse comparing with monovalent ions [32]. Jordan et al. found that the attractive interaction is induced by the binding of multivalent ions, most likely due to ion bridging between protein molecules [33]. Upon the addition of salt, the electrostatic interactions between charged groups on the polymer backbone are screened and the chains favor to collapse with salt concentration increase. In the presence of excess multivalent salt in the solution, it was also observed that chains re-expand when the salt concentration increases to a higher value [34].

Up to now, there have been only a few studies involving a semiflexible PE in multivalent salt solution, and the semiflexible PE chain is linear [26,35]. The dynamics of PEs with different architectures other than linear counterparts is very important because these PEs play unique roles in nature, and also show special properties in many aspects. Biology is full of natural circular polymers that bear charge, such as cyclic deoxyribonucleic acid and circular PEs, which are discovered in viruses and bacteria, etc. The cyclic or ring polymers have a topologically interesting structure with no chain ends, and their dimensional and various physical properties have attracted much attention. In the paper, we use molecular dynamics simulations to study the effect of the tetravalent salt concentration on the conformation of semiflexible ring polyelectrolyte in dilute and semi-dilute salt solution regimes. To elucidate the effect of the chain rigidity and salt concentration on the conformation of semiflexible ring PE, we have performed simulations of PE with different bending rigidities in tetravalent salt solution.

## 2. Results and Discussion

### 2.1. Equilibrium Phases of the Semiflexible Ring Polyelectrolyte

Before discussing our study results, three important parameters are stated at first. The Bjerrum length is set to 3σ = 7.2 Å in this work, according to the Ref. [36]. The persistence length *l*_p_ is taken as the basic characteristic of polymer flexibility. In this work, we use the bending energy *b* to characterize the stiffness of a polymer. Based on the method described in Ref. [37], the relation between *b* and *l*_p_ is established, as shown in Figure 1A. The tetravalent cations play a decisive role in the folding process of the ring PE. Knowing how many cations there are in solution of specific salt concentration, it is very important to understand the phase transition. Each salt molecule contains one tetravalent cation and four anions. Therefore, the number of tetravalent cations is equal to the number of salt molecules *N_s_*_alt_ at any salt concentration. Figure 1B shows how many salt molecules there are at different *C_S_*, based on the equation *N*_salt_ = *C*_S_*VN*_A_ (where *V* is the volume of the simulation box, and *N*_A_ is the Avogro constant).

The model of semiflexible ring PE is coarse-grained bead-spring. The bending potential is introduced to characterize the rigidity of the charged biomacromolecule and charged synthetic macromolecule. To emphasize the basic property, other properties are ignored. Although there is no twisting potential in the model of the ring PE, as the bending energy *b* increases, the twisting interaction enhances accordingly. For clarity, only semiflexible ring PE chain is shown in the snapshots. Figure 2 shows the conformations of semiflexible ring PE chain of different rigidities at salt concentration *C_S_* = 1.78 × 10^−3^ M (number of salt molecules *N*_salt_ = 50). It can be observed that bending energy *b* has an obvious impact on the conformation of the semiflexible ring PE chain. The ring PE chain folds into compact globule at *C_S_* = 1.78 × 10^−3^ M for *b* = 0. The ring PE chain is flexible for *b* = 0. In the presence of tetravalent salt cations, there exist strong electrostatic attractions between the monomer beads and the tetravalent salt cations that can effectively screen the charges on the ring PE backbones. These cations play the role of a bridge between the monomers. When the salt solution is dilute, the cation-bridging effort is dominant for the flexible ring PE. Therefore, the chain assumes compact globule. As *b* increases, the ring PE chain folds into toroid at *b* = 20 (the persistence length *l_p_* = 14.08) and *b* = 40 (*l_p_* = 28.05), and the number of torus rings *N*_ring_ decreases. When *b* is larger than 1.0, the ring PE chain is of rigidity, and its rigidity increases with *b* increase. The competition among the bending energy of PE chain, cation-bridging interaction, and entropy is very fierce. However, none of them plays dominant role at *b* = 20 and *b* = 40. The ring PE chain condenses into toroid. As *b* increases further, the ring PE chain forms spindle with two racquet heads at *b* = 60 (*l_p_* = 42.04), 80 (*l_p_* = 56.08) and 120 (*l_p_* = 84.14). The bending energy begins to hold the upper hand, but does not take the dominant role under these circumstances.

For the sake of analyzing the structure of a chain, one needs a method that will let us know the monomer contact distribution. It can be achieved by generating so-called contact maps, which are essentially a representation of 3D connectivity in a two-dimensional map. We regarded a monomer *i* and a monomer *j* in contact when they are found within the radius of 2.5 [38] and |*i* − *j*| is larger than 2. Figure 3A shows the contact maps of the corresponding representations of the ring PE’s conformations from Figure 2. In addition, the average cosine $<\cos\theta_{i,j}>$ between the backbone bond vector ${\vec{u}}_{i}$ and ${\vec{u}}_{j}$ as a function of the distance along the chain backbone |*i* − *j*| is introduced. Figure 3B shows the $<\cos\theta_{i,j}>$ of the corresponding representations of the ring PE’s conformation from Figure 2. By combining them, the conformations of ring PE can be extremely correctly characterized. Both the contact map and $<\cos\theta_{i,j}>$ are totally disordered for *b* = 0. The contacts give rise to a positive slope in the contact map, and $<\cos\theta_{i,j}>$ assumes periodicity for *b* = 20 and 40. In addition, the number of straight lines is one fewer than *N*_ring_, while the number of periods of $<\cos\theta_{i,j}>$ is equal to *N*_ring_. The contacts give rise to negative slope in the contact map, and the curve of $<\cos\theta_{i,j}>$ is V-shaped for *b* = 60, 80 and 120.

Next, we study the conformation of the ring PE at different salt concentrations *C_S_* for *b* = 60, as shown in Figure 4. The conformation of the ring PE is loop-like at *C_S_* = 3.56 × 10^−4^ M (*N*_salt_ = 10). The salt solution is very dilute at *C_S_* = 3.56 × 10^−4^ M. There are only ten tetravalent cations in the solution, i.e., the Debye screening length is very large. The cation bridge does not affect chain conformation, while the rigidity of the ring PE is moderate. Therefore, the bending energy plays dominant role. As the *C_S_* increases, the ring PE folds into two-racquet-head spindle at *C_S_* = 1.24 × 10^−3^ M (*N*_salt_ = 35) and 1.78 × 10^−3^ M (*N*_salt_ = 50). More tetravalent salt is added into solution. The cation -bridge begins to exert an influence, but its influence is not large enough to drive the ring PE bend. Therefore, the ring PE chain forms two-racquet-head spindle. Upon the continued addition of salt, the conformation of ring PE becomes toroid at *C_S_* = 1.96 × 10^−3^ M (*N*_salt_ = 55) and 8.54 × 10^−2^ M (*N*_salt_ = 2400). There are enough tetravalent cation to screen the electrostatic repulsion between monomers. The competition among the bending energy, cation bridge, and entropy is very fierce. The cations cause a stronger bridging of monomers of the ring PE. Forming toroid is beneficial to entropy increase and energy reduction. As more salt is added, the ring PE folds into the two-racquet-head spindle again at *C_S_* = 8.90 × 10^−2^ M (*N*_salt_ = 2500) and 1.21 × 10^−1^ M (*N*_salt_ = 3400), and the size of the racquet head increases with the increase of *C_S_*. Upon the addition of the multivalent salt, the Debye screening length decreases with *C_S_* increase. The bending energy takes the upper hand. At *C_S_* = 1.24 × 10^−1^ M (*N*_salt_ = 3600), the conformation of ring PE becomes loop again. In the presence of an excess of multivalent salt present in the solution, the Debye screening length is very small. The influence of cation bridge is very trivial. The bending energy plays the dominant role. The semiflexible ring PE re-expands with a further increase of the salt concentration.

The Figure 5 shows the contact maps and $<\cos\theta_{i,j}>$ of the ring PE in the corresponding snapshots of Figure 4. If the conformation is loop-like, there are no contacts in the contact maps for *C_S_* = 3.56 × 10^−4^ M and at *C_S_* = 1.24 × 10^−1^ M, as shown in Figure 5A, and the curve of $<\cos\theta_{i,j}>$ is V-shaped, as shown in Figure 5B. If the conformation is two-racquet-head spindle, the contacts give rise to negative slope in the contact map for *C_S_* = 1.24 × 10^−3^ M, 1.78 × 10^−3^ M, 8.90 × 10^−3^ M and 1.21 × 10^−1^ M and curve of $<\cos\theta_{i,j}>$ is V-shaped. If the conformation is toroid, the contacts give rise to positive slope in the contact map for *C_S_* = 1.96 × 10^−3^ M and 8.54 × 10^−3^ M, and $<\cos\theta_{i,j}>$ assumes periodicity. By combining the contact maps and $<\cos\theta_{i,j}>$, the five phases of the ring PE can be extremely correctly characterized.

### 2.2. Statistical Properties of the Ring Polyelectrolyte

To discuss the shape of the ring PE, two characteristic quantities are introduced to characterize its shape. The mean radius of gyration $<R_{g}^{2}{>}^{1/2}$ scaled by its equilibrium value is calculated to analyze the conformation of the ring PE at different *b* and *C_S_*. The value of the $<R_{g}^{2}{>}^{1/2}$ can be used as a reasonable dividing boundary among different conformations. To better discriminate the conformations of the ring PE, its average shape factor <δ> is studied. The <δ> is calculated by the method described in Refs. [39,40]. It varies between 0 (sphere) and 1 (rod).

Figure 6 shows that the $<R_{g}^{2}{>}^{1/2}$ and <δ> respond to the salt concentration for *b* = 0, 10, 60, and 120. By combining them, the phase transition can be discriminated accurately. Upon the addition of salt, the conformation of the flexible ring PE shifts from loop to compact globule firstly, and then becomes coil for *b* = 0. As the ring PE is of rigidity, different phase transitions are observed. The conformation of the ring PE shifts from loop to bended spindle firstly, and then becomes compact toroid, finally becomes coil with the increase of *C_S_* for *b* = 10. As the rigidity enhances further, more phases are observed. The conformation of ring PE shifts from loop to two-racquet-head spindle firstly, subsequently becomes toroid, then converts into two-racquet-head spindle again, finally assumes loop-like again for *b* = 60 and 120. As *b* increases, the salt concentration of the phase transition point increases firstly, then decreases. Figure 7 shows that $<R_{g}^{2}{>}^{1/2}$ and <δ> respond to the bending energy *b* at different *C_S_*. The flexible ring PE folds into compact globule at *C_S_* = 1.78 × 10^−3^ M, 7.12 × 10^−3^ M, and 5.69 × 10^−2^ M for *b* = 0. The salt solution is dilute at *C_S_* = 1.78 × 10^−3^ M. Only scores of tetravalent cations are added. The screening Debye length is very large. Therefore, the influence of cation bridge is weak. The ring PE folds into toroid in the range of *b* = 10~40, and its conformation becomes two-racquet-head spindle in the range of *b* = 60~120. As *C_S_* increases to *C_S_* = 7.12 × 10^−3^ M, more tetravalent cations are added, and the screening Debye length is moderate. The influence of cation-bridge is enhanced. The ring PE folds into toroid in the range of *b* = 10–100, and its conformation becomes two racquet head spindle at *b* = 120. The screening debye length is short at *C_S_* = 5.69 × 10^−2^ M. The influence of cation bridge becomes weak again. The ring PE condenses into toroid in the range of *b* = 10~60, then folds into two-racquet-head spindle at *b* = 80, finally its conformation becomes loop in the range of *b* = 100~120. The radial density distributions *P*(*r_m_*) of monomers relative to the center of mass of the ring PE further verify the results (see the Appendix A).

The toroid is the most important structure. Next, we discuss it in detail. When the tetravalent cations begin to condense onto the ring PE backbone [41], monovalent counterions move away from the ring PE [23]. These condensed tetravalent cations may also cause localized twisting of the ring PE, which can facilitate collapse [42]. Due to monovalent counterions far away from ring PE, the translational entropy counterions is produced, which drives the ring PE twist further to become toroid. Therefore, it exerts an important effect on the folding of the ring PE [43]. When the distance from tetravalent cations to the ring PE backbone is less than the equilibrium bond length, the tetravalent is considered to be condensed. Figure 8A shows that the number of condensed tetravalent cation *N*_c_ responds to *C_S_*. The changing trend of *N_c_* is in agreement with that of the conformation. In the regime of small *C_S_* and large *C_S_*, the *N_c_* is small, which is one of reasons for the ring PE folding into loop. The *N*_c_ almost keeps constant in the regime of the toroid. It means that the number of torus ring *N*_ring_ is almost the same at different *C_S_* in the regime of toroid for the same *b*. In addition, it can be observed that the total charge of the ring PE is neutralized in the regime of the toroid. The ring PE collapse is also determined by the total charge neutralization of the ring PE. Figure 8B shows that *N_ring_* and $<R_{g}^{2}{>}^{1/2}$ respond to *b*. It can be observed that the *N_ring_* decreases firstly, and then keeps constant with *b* increase. The largest *N_ring_* is 5.8 at *b* = 10. The rigidity of the ring PE is very small at *b* = 10, therefore, the ring PE can be twisted easily. With *b* increase from 10 to 60, *N_ring_* decreases to 2. The ring PE becomes more and more rigid, i.e., it is more and more difficult for ring PE to twist. *N_ring_* keeps constant when *b* is larger than or equal to 60. The range of *C_S_* for toroid becomes very small at *b* = 120. The changing trend of $<R_{g}^{2}{>}^{1/2}$ is in agreement with that of *N_ring_*. It further verifies that the *N_ring_* decreases with *b* increase.

### 2.3. Phase Diagram of the Ring Polyelectrolyte

Summing up all results, we have constructed the diagram of conformations of the ring PE in terms of bending energy *b* and salt concentration *C_S_*. It is shown in Figure 9 where the following stable states can be distinguished depending on *b* and *C_S_*: loop, two-racquet-head spindle, toroid, globule, and coil. For *b* = 0, the flexible ring PE folds into loop firstly, then its conformation becomes compact globule, finally it extends into coil upon the addition of salt. As the ring PE is of stiffness, the phase transition of the semiflexible ring PE is different from that of the flexible ring polyelectrolyte. For *b* = 10, the conformation of the ring PE shifts from loop to bended spindle firstly, then it condenses into toroid, finally it folds into coil with *C_S_* increase. When the rigidity of ring PE increases, the ring PE experiences four phase transitions with *C_S_* increase. Upon the addition of salt, the phase shifts from loop to spindle at first, secondly shifts from spindle to toroid, thirdly shifts from toroid to spindle, finally shifts from spindle to loop. The larger the *b* is, the higher the *C_S_* of first phase transition point is. The *C_S_* of the second phase transition point also increases. The salt concentration range of toroid becomes more and more narrow with *b* increase. The *C_S_* of the latter two phase transition points decreases with *b* increase.

The phase is determined by the competition among the bending energy, cation bridge, and entropy. Upon the addition of salt, the Debye screening length decreases. When the salt solution is very dilute, the Debye screening length is very large and *N_c_* is small. The influence of the cation bridge is trivial. The bending energy plays a dominant role. Therefore, the semiflexible ring PE folds into loop. As *C_S_* increases, the cation bridge takes effect. There are enough condensed cations to drive the ring PE collapse. The phase is determined by the main competition between the cation-bridge and bending energy. The cation bridge takes the upper hand. The semiflexible ring PE folds into spindle. As *C_S_* increases further, the Debye screening length is moderate. The cation bridge plays an important role. The toroidal structure is helpful to increase entropy of the system. The competition among the bending energy, cation bridge, and entropy is very fierce. The semiflexible ring PE condenses into toroid. As *C_S_* increases to a certain value, the Debye screening length becomess short, i.e., the influence of the cation bridge weakens, and the condensed cations are too few to drive the ring PE collapse. The semiflexible PE folds into spindle again. As *C_S_* is very large, the Debye screening length is very short. The influence of bending energy becomes dominant. The conformation of the semiflexible ring PE becomes loop-like again. The results of Golan show that toroid and rod coexist [44]. Their DNA molecules fold several times into very shorter rods. Our ring PE is not long enough to fold several times. It just draws close to become a rod. The barriers for a fully condensed spindle to convert into a toroid are high. Unfolding in poor solvent conditions is highly unfavorable [38]. Therefore, we do not observe the coexistence of toroid and rod. In our previous work, we only observed that the semiflexible ring PE folds from loop to spindle at first, then to toroid with the increase of strength of electrostatical interaction [45].

## 3. Model and Methods

### 3.1. The Coarse-Grained Model of the Semiflexible Ring Polyelectrolyte in Multivalent Salt Solution

Our system is made up of one semiflexible ring PE chain, counterions, and salt contained in a periodic cubic box of side length L = 150 σ, as shown in Figure 10. The periodic boundary conditions of a cubic box are imposed along all three directions (x, y, and z). The semiflexible ring PE chain is generated by a coarse-grained bead-spring model, where each bead carries a negative unit charge −e. The chain contains N = 200 monomers. To make sure that the system is electrically neutral, N_c_ = 200 counterions is explicitly introduced, where each counterion possesses a positive unit charge e. To elucidate the influence of multivalent salt on the folding behaviors of the semiflexible ring PE, tetravalent salt are explicitly added into the solution where each salt molecule decomposes into one cation of 4e and four anions of −e. The tetravalent salt cations, the monovalent salt anions, and the monovalent counterions are modeled as charged beads whose diameters, σ and mass m are the same.

### 3.2. Molecular Dynamics Simulations

Simulations are carried out under NVT conditions in a periodic cubic box using the open-source software LAMMPS (version GPLv2) [46]. There are four types of interactions in the simulations. The first one is the short-range repulsive potential which accounts for excluded volume interaction among all the beads, including monomers of semiflexible ring PE, counterions, salt cations, and salt anions. It is modeled by the Lennard-Jones (LJ) potential [47],
(1)$$U_{LJ}(r_{ij})=4\varepsilon \left[{\left(\frac{\sigma }{r_{ij}}\right)}^{12}-{\left(\frac{\sigma }{r_{ij}}\right)}^{6}\right]$$
where r_ij_ is the distance between the center of particle i and the center of particle j, and σ is the inter-particle distance at which the potential is zero. Both ε and σ (in reduced units) are set to ε = 1.0 and σ = 1.0 for all pairs of beads. The LJ potential goes to zero beyond a cutoff distance r_c_ = σ, such that the excluded volume interaction between any two beads is purely repulsive [47,48].

The second one is the bond connectivity interaction. Adjacent beads along the semiflexible ring PE chain are connected by harmonic springs [45],
(2)$$U_{bond}(r_{ij})=\frac{1}{2}k{\left(r_{ij}-r_{0}\right)}^{2}$$
where k is the spring constant and r_0_ is the equilibrium bond length. We set r_0_ = 1.12 σ and k = 500.0.

The third one is a harmonic angle potential, which characterizes the ring PE chain stiffness [47]
(3)U=b(1+cosθ)
where θ is the angle between two consecutive bonds. and b is the bending energy. b is regarded as a penalty for successive bonds deviating from a straight arrangement, i.e., the chain rigidity can be controlled by b. In addition, b is in the units of k_B_T, where k_B_ is the Boltzmann constant and T is the absolute temperature.

We add N_i_ = C_S_VN_A_ tetravalent salt to simulate a system with molar concentration C_S_, where V = L^3^ is the volume of the cubic box. The Debye length of the system [49], including the salt ions and counterions, is then
(4)$$\lambda_{D}={\left[4\pi \lambda_{B}(n_{sa}z_{sa}^{2}+n_{sc}z_{sc}^{2}+n_{c}z_{c}^{2})\right]}^{-1/2}$$
where $\lambda_{B}=e^{2}/(4\pi \varepsilon_{0}\varepsilon_{rs}k_{B}T)$ is the Bjerrum length, with $\varepsilon_{0}$ and $\varepsilon_{rs}$ being the vacuum and the relative permittivity, respectively. $z_{sa}$, $z_{sc}$, and $z_{c}$ are the valency of salt anion, salt cation, counterion, respectively.$n_{sa}=N_{sa}/L^{3}$,$n_{sc}=N_{sc}/L^{3}$, and $n_{c}=N_{c}/L^{3}$ are ion density, where $N_{sa}$, $N_{sc}$ and $N_{c}$ are the number of salt cations, salt anions, and counterions, respectively.

The fourth one is Debye–Hückel potential between any two charged beads q_i_ and q_j_ in the system [50]:(5)$$U_{C}(r)=\{\begin{array}{c} \frac{\lambda_{B}q_{i}q_{j}}{r}\exp(-r/\lambda_{D}),\;r<r_{cut}\\ 0,\;\;\;\;\;\;\;r>r_{cut} \end{array}$$

The Bjerrum length is set to λ_B_ = 3.0 σ [36], while r_cut_ = 5 λ_D_ [50].

The simulations are performed using LAMMPS package [51]. In this study, all the physical quantities are represented in the standard LJ units (i.e., mass m, length σ and energy k_B_T). A Langevin thermostat [52] is used to maintain a constant temperature (T = 1.0) of the system, and we set the friction coefficient ξ=m/τ where $\tau =\sqrt{m\sigma^{2}/\varepsilon }$ is the time unit of the system. We set k_B_ = 1 and m = 1 for convenience. The time step for integrating equations of motion is chosen as 0.001. We start from a randomly distributed configuration. The system relaxes with 10^7^ steps to reach equilibration. Then the production run is carried out by performing 10^7^ steps to sample and data are saved every 10^4^ timesteps, resulting in 1000 regularly time-spaced snapshots, and 10 independent runs are carried out.

## 4. Conclusions

In this work, we have investigated the effects of salt concentration, and intrinsic chain stiffness on the conformation of the ring PE in the tetravalent salt solution, using the molecular dynamics method. To emphasize the stiffness of the ring PE, other properties are ignored. The stiffness of the ring PE is introduced via a potential depending on the angle between two adjacent bond vectors with the stiffness parameter *b*. Throughout our calculations, the *b* varies from *b* = 0 (freely jointed chain) to *b* = 120. The salt concentrations *C_S_* change in the range of 3.56 × 10^−4^ M ≤ *C_S_* ≤ 2.49 × 10^−1^ M. The influence of *C_S_* and the role of chain stiffness on the ring PE’s conformation are discussed in detail.

The morphology of the ring PE can be controlled and the self-assembly process can be tuned by altering *b* and *C_S_*. The phase transition of the semiflexible ring PE is totally different from that of the flexible ring PE. Upon addition of salt, more abundant conformations were observed for the semiflexible ring PE. The bending energy produces an important effect on the conformation of the ring PE. For *b* = 10, there are three phase transitions. The conformation of the ring PE shifts from loop to spindle firstly, then ring PE condenses into toroid, finally arranges into coil, upon the addition of salt. When the stiffness of the ring PE increases, there are four phase transitions. The latter two are phase transitions from toroid to spindle, and from spindle to loop, respectively. The *C_S_* of the phase transition point increases with *b* increase for the former two phase transition, while it decreases with *b* increases for latter two phase transitions. The structural changes are found to be guided by the strong interaction of the cations with semiflexible ring PE. The Debye screening length is very large in dilute solution. The influence of cation bridge is trivial. The semiflexible ring PE folds into loop. Initially, an increase in salt concentration facilitates the structural distortion of ring PE and helps to bring it close. The semiflexible ring PE arranges into spindle. As *C_S_* increases further, the Debye screening length is moderate. The cation bridge plays an important role. The main competition comes from cation-bridge and entropy. The semiflexible ring PE condenses into toroid. As *C_S_* increases to certain value, the debye screening length is short. The influence of cation-bridge weakens. The main competition comes from cation-bridge and bending energy. The conformation of the semiflexible ring PE becomes spindle again. As *C_S_* is very large, the Debye screening length is very small. The influence of cation bridge is trivial. The semiflexible ring PE folds into loop again.

Our study presented here goes further to investigate in detail the microstructure of a ring PE which gives us a thorough vision of the behavior of polyelectrolytes in the presence of multivalent salt. We believe that these results are beneficial to improve our understanding of the conformational behavior of charged biopolymers.

## Figures and Tables

**Figure 1 ijms-25-08268-f001:**
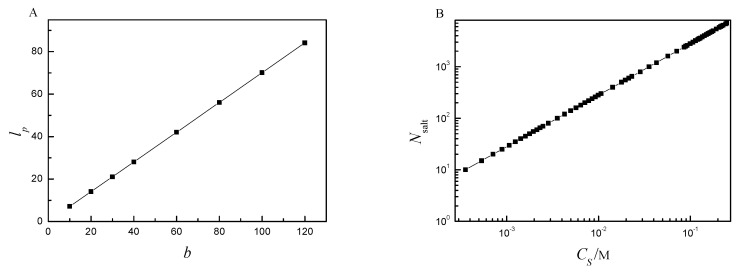
(**A**)The persistence length *l*_p_ vs. bending energy *b*; (**B**) the number of salt molecules *N*_salt_ in solution vs. salt concentration *C_S_*.

**Figure 2 ijms-25-08268-f002:**
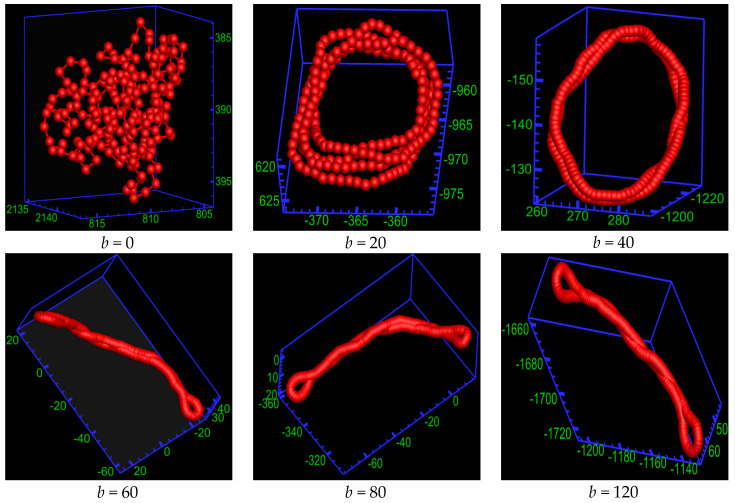
Simulation snapshots of semiflexible ring polyelectrolytes of different rigidity at salt concentration *C_S_* = 1.78 × 10^−3^ M. For the clarity of the pictures, only the semiflexible ring polyelectrolyte is plotted.

**Figure 3 ijms-25-08268-f003:**
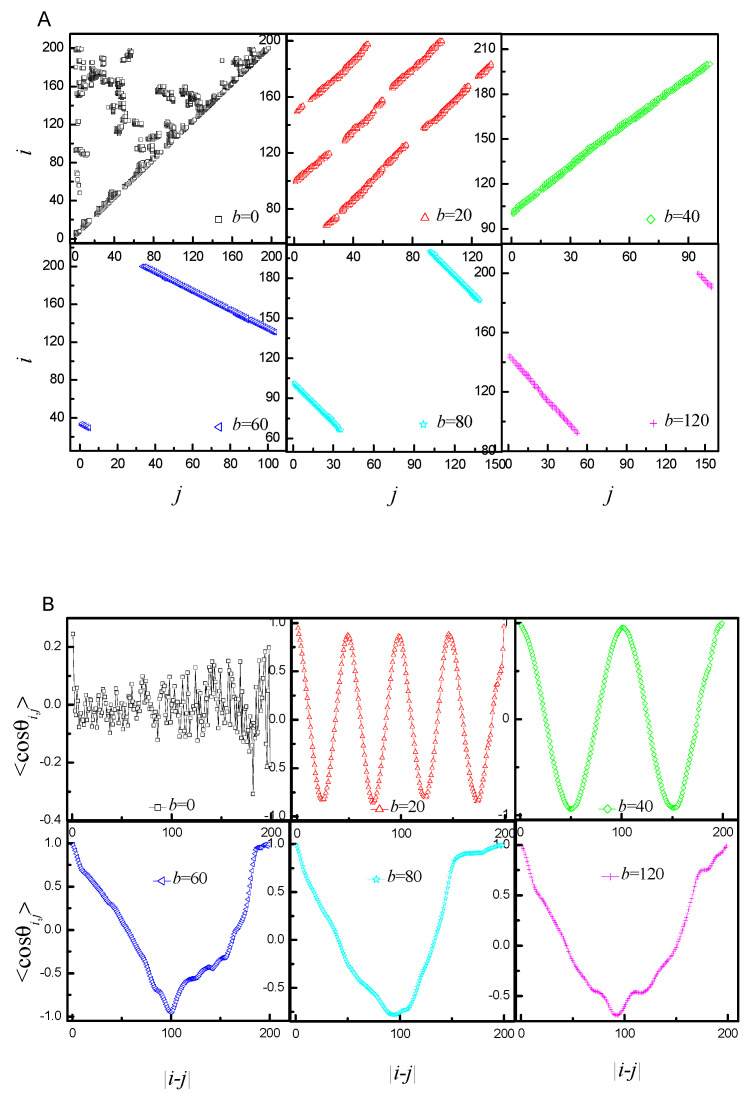
(**A**) Corresponding representations of the ring polyelectrolyte’s conformation from Figure 1 in the contact map space; (**B**) the average cosine between bonds vs. the distance along the ring PE in the corresponding snapshots of Figure 1.

**Figure 4 ijms-25-08268-f004:**
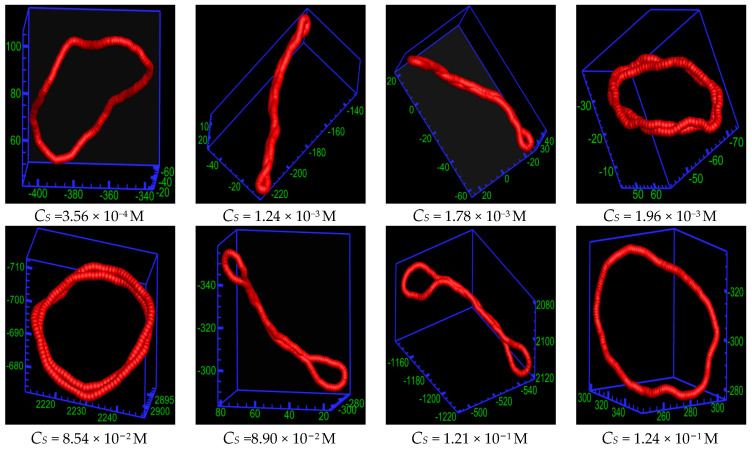
Simulation snapshots of the semiflexible ring polyelectrolyte of *b*=60 at different salt concentrations. For the clarity of the pictures, only the semiflexible ring polyelectrolyte is plotted.

**Figure 5 ijms-25-08268-f005:**
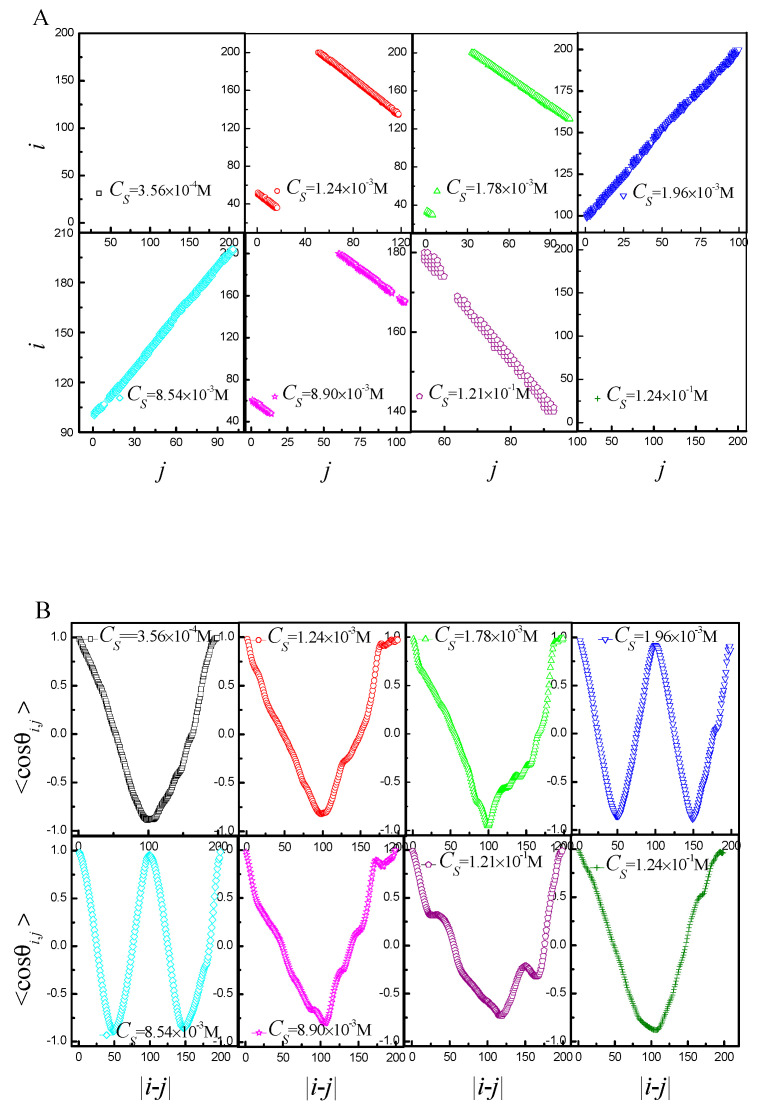
(**A**) Corresponding representations of the ring polyelectrolyte’s conformation from Figure 4 in the contact map space; (**B**) the average cosine between bonds vs. the distance along the ring PE in the corresponding snapshots of Figure 3.

**Figure 6 ijms-25-08268-f006:**
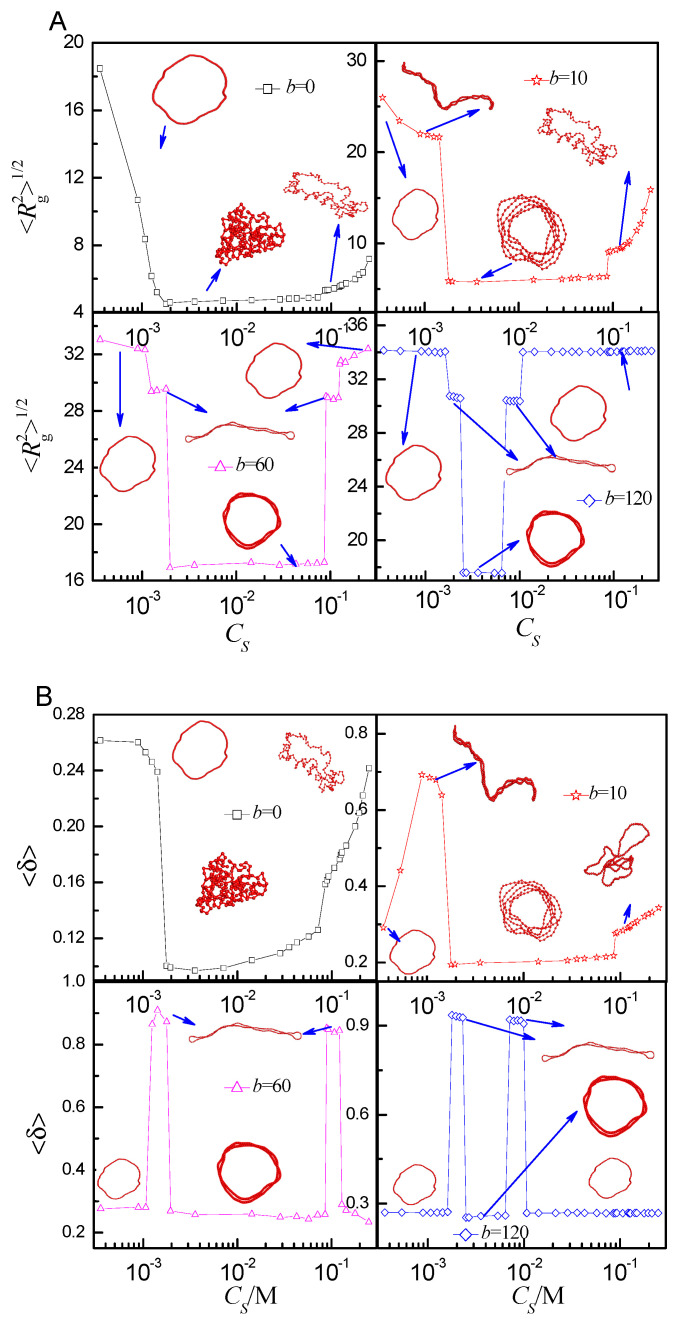
(**A**) The mean radius of gyration $<R_{g}^{2}{>}^{1/2}$ as a function of salt concentration *C_S_* for different bending energy *b*; (**B**) Shape factor <δ> as a function of *C_S_* for different *b*.

**Figure 7 ijms-25-08268-f007:**
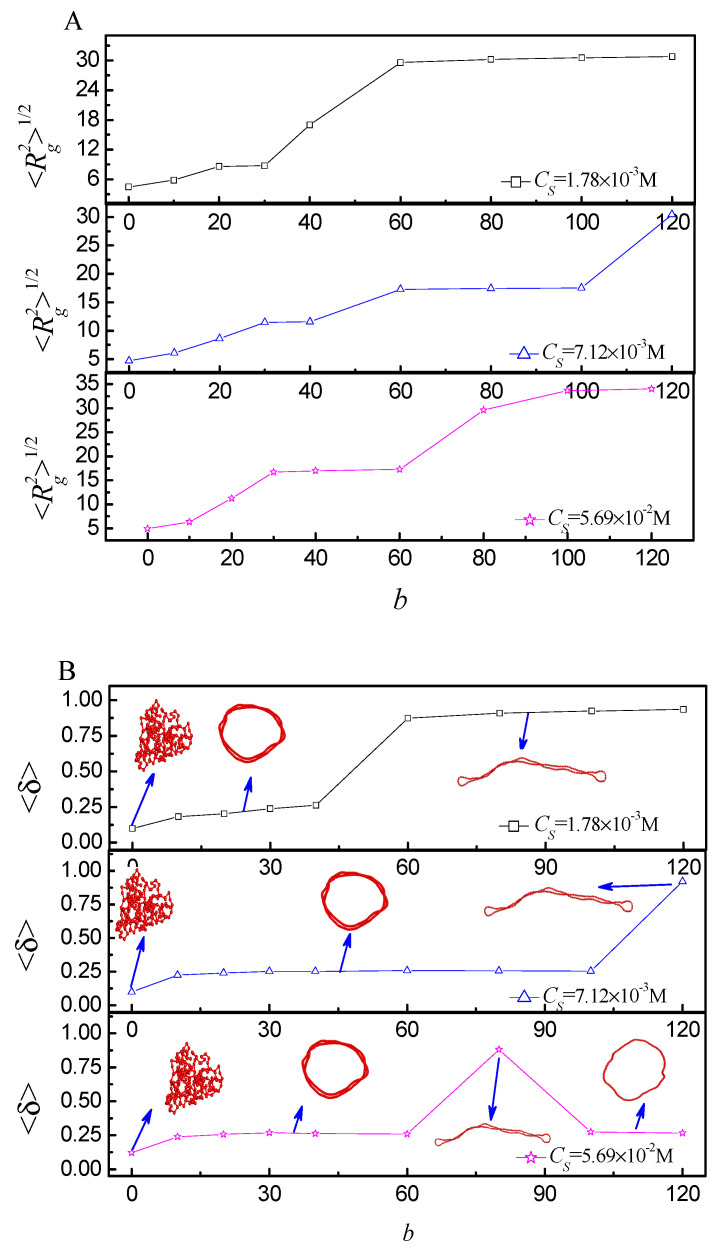
(**A**) The mean radius of gyration $<R_{g}^{2}{>}^{1/2}$ as a function of different bending energy *b* at different salt concentrations *C_S_*; (**B**) the shape factor < δ> as a function of *b* at different *C_S_*.

**Figure 8 ijms-25-08268-f008:**
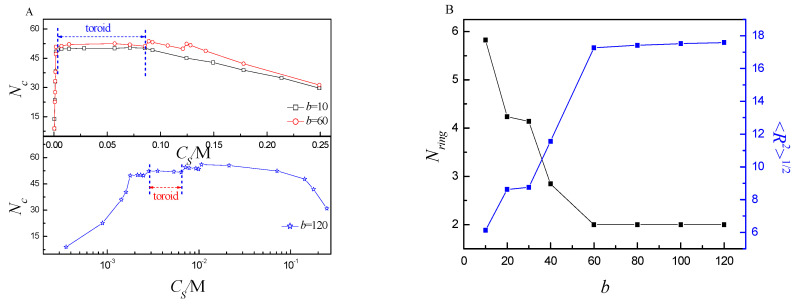
(**A**) The number of torus rings *N_ring_* and the mean radius of gyration $<R_{g}^{2}{>}^{1/2}$ as a function of different bending energy *b*; (**B**) the number of tetravalent cations *N_c_* binding to ring PE as a function of *C_S_* at different *b*.

**Figure 9 ijms-25-08268-f009:**
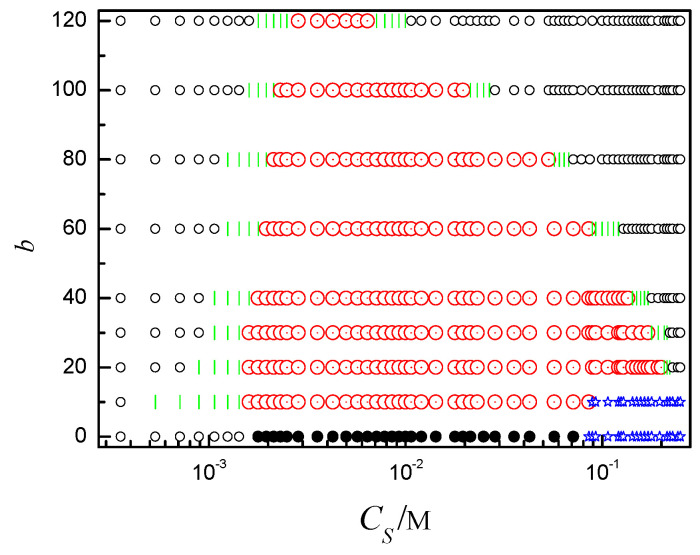
The ring polyelectrolyte state diagram in the values of bending energy *b* and salt concentration *C_S_*. Black ○ denotes loop, green | denotes two racquet head spindle, black ● denotes globule, blue ☆ denotes coil, red ☉ denotes toroid.

**Figure 10 ijms-25-08268-f010:**
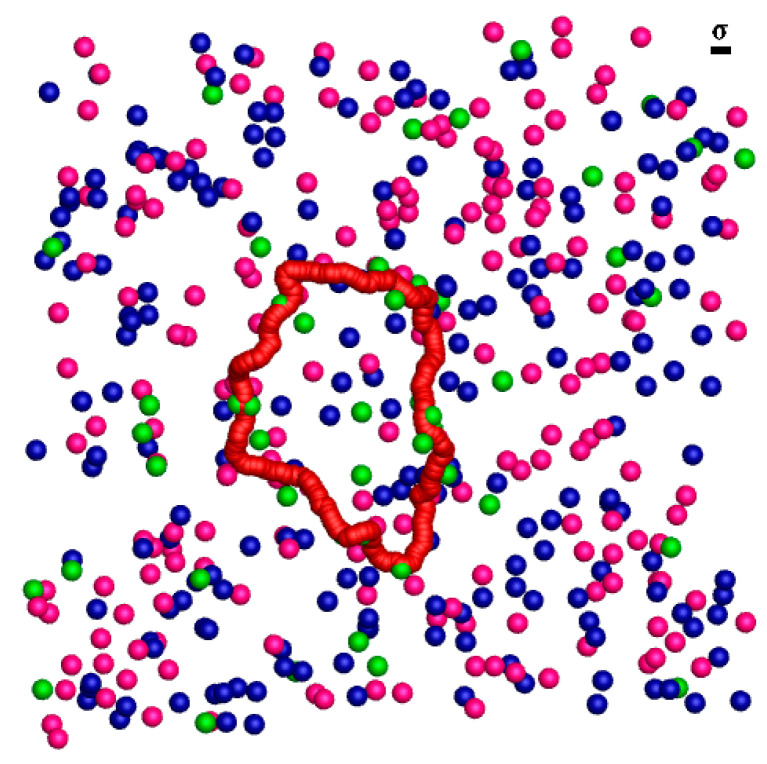
Snapshots of a single semiflexible ring polyelectrolyte in the tetravalent salt solution. The semiflexible ring polyelectrolyte is shown in red, the counterions are shown in blue, the monovalent salt anions are shown in magenta, and tetravalent salt cations are shown in green.

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
