# Peer review of "Conformational Transition of Semiflexible Ring Polyelectrolyte in Tetravalent Salt Solutions: A Simple Numerical Modeling without the Effect of Twisting"

_ijms, 2024, doi:10.3390/ijms25158268_

Round 1
Reviewer 1 Report
Comments and Suggestions for Authors
The reviewer regards that there is no significant problem in the linguistic property.
Author Response
The model of semiflexible ring PE is coarse-grained bead-spring. The twisting interactions/potentials is introduced to characterize the rigidity of charged biomacromolecule and charged synthetic macromolecule. In order to emphasize the basic property, other properties are ignored in our study work. We have calculated the the number of torus ring and the number of condensed tetravalent cations in the revised manuscript, and discuss them in detail.

Reviewer 2 Report
Comments and Suggestions for Authors
In this work, the effects of salt concentration, and intrinsic chain stiffness on the conformation of ring PE in tetravalent salt solution is investigated by molecular dynamics. The proposed work is interesting but serious concern need to be addressed :
- The authors need to explain the novelty of this work compared with their previous article (https://doi.org/10.1007/s10118-022-2842-x Chinese J. Polym. Sci. 2023, 41, 448–458)
- It is not clear how the solvent was taken into account in the molecular dynamics. Author should specify it as solvent is a critical parameter governing the behaviour of polyelectrolytes
- Assuming that an implicit solvent was used for modelization, could the authors comment on the possibility to reproduce their results using explicit solvent ?
- To better illustrate the significance of the salt valency in polyelectrolyte materials, the following experimental systems should be cited/described: J. Mater. Chem. A, 2020, 8, 17713-17724 (polyelectrolyte gels vs salt valency), LbL films (Langmuir 2012, 28, 45, 15831–15843)
- Figures 7,8,9 are unreadable and should be redrawn
- Figures 6 and 7 have less significance and should be moved to SI
Author Response
In this work, the effects of salt concentration, and intrinsic chain stiffness on the conformation of ring PE in tetravalent salt solution is investigated by molecular dynamics. The proposed work is interesting but serious concern need to be addressed :
- The authors need to explain the novelty of this work compared with their previous article (https://doi.org/10.1007/s10118-022-2842-x Chinese J. Polym. Sci. 2023, 41, 448–458)
Answer: We have discussed the difference between the two study works and novelty of this work in the revised manuscript.
- It is not clear how the solvent was taken into account in the molecular dynamics. Author should specify it as solvent is a critical parameter governing the behaviour of polyelectrolytes
Answer: In the molecular dynamics, the friction coefficient is used to characterize the solvent. We have stated it in the revised manuscript.
- Assuming that an implicit solvent was used for modelization, could the authors comment on the possibility to reproduce their results using explicit solvent ?
Answer: The simulation results are in agreement with the experimental results in most cases. I think we still can reproduce our results in explicit solvent.
- To better illustrate the significance of the salt valency in polyelectrolyte materials, the following experimental systems should be cited/described: J. Mater. Chem. A, 2020, 8, 17713-17724 (polyelectrolyte gels vs salt valency), LbL films (Langmuir 2012, 28, 45, 15831–15843)
Answer: We have cited them in the revised manuscript.
- Figures 7,8,9 are unreadable and should be redrawn
Answer: We have redrawn them in the revised manuscript.
- Figures 6 and 7 have less significance and should be moved to SI
Answer: We have moved them to SI.

Round 2
Reviewer 1 Report
Comments and Suggestions for Authors
With respect to the twisting interaction/potential:
The authors newly added a simple explanation just below Figure 1:
“The model of semiflexible ring PE is coarse-grained bead-spring. The twisting interactions/potentials is introduced to characterize the rigidity of charged biomacromolecule and charged synthetic macromolecule.”
Unfortunately, there is no additional description concerning the twisting interaction in the revised version.
In the explanation of “3.2. Molecular Dynamic Simulation”, the authors explained the interactions adapted in the present study; 1) Excluded volume interaction (Lennard-Jones potential), 2) Bond connectivity between adjacent beads, 3) Harmonic bending potential, (bending potential), and Coulombic interactions (Debye-Huckel) Thus, it is quite obvious that authors completely ignored the effect of twisting interaction, which is generally very important for ring-shaped semiflexible polyelectrolytes, such as double-stranded DNA above the size of kbp. As the reviewer, it is quite strange why the authors added the statement of “The twisting interactions/potentials is introduced” in the revised version, against the actual model adapted for the simulation. Bending and twisting are physically different subjects each other as has been well explained in usual textbook of polymer physics.
With respect to the effect of counter ion condensation:
In the revised version, the authors newly added the following interpretation by adding references [39], [23].
The tetravalent cations can condense onto the ring PE backbone[39], while they drive away the monovalent counterions[23].
Unfortunately, this statement did never play the role as a reasonable response in the revision. If the authors carefully read and check the related article concerning the effect of “counter ion condensation” on the “condensation of polyelectrolyte”, they may recognize the very important and nonnegligible effect of the competition between monovalent and multivalent counter ions through the “counter ion condensation”. In other words, changes in the translational entropy of monovalent cations play the dominant role in the stability of the condensed polyelectrolytes. The reviewer would like to suggest the authors to learn the past representative studies concerning the effect of counter ions, as exemplified in the following.
H. Schiessel, et al., Counterion-Condensation-Induced Collapse of Highly Charged Polyelectrolytes. Macromol., 31, 7953(1998). Cited by 275.
V. A. Bloomfield, DNA Condensation by Multivalent Cations, Biopolymers, 44, 269(1997). Cited by 1399.
In the present study, the authors adapted the Debye-Huckel approximation. The validity of counter ion condensation hypothesis indicates the break-down of the framework of Debye-Huckel, owe to the higher-order correlation among ionic species. Thus, it is necessary for the author to take into consideration of the effect of the drastic change on the manner of counter ion condensation accompanied by the folding transition of polyelectrolyte chains.
Minor comments:
On Figures 1,3,9: For these pictures, the length scales are different each other. It may be better to show the scale bar with the unit of σ.
On Figures 2A and 4A: Why are the horizontal and vertical scales different for each diagram?
On Figures 5-8: The authors showed various plots, indicating that there are no bistability, or coexisting of different states, for the simulation with the same parameters. This seems unrealistic. Past experimental and theoretical studies indicated that different morphology coexists in many situations. For example, coexistence of toroid and rod is well known for the folded compact DNA, as exemplified in the following article.
R. Golan, et al., DNA Toroids: Stages in Condensation, Biochem., 38, 14069(1999). Cited by 273
Comments on the Quality of English LanguageLinguistic quality seems to be not so bad.
Author Response
The authors newly added a simple explanation just below Figure 1:
“The model of semiflexible ring PE is coarse-grained bead-spring. The twisting interactions/potentials is introduced to characterize the rigidity of charged biomacromolecule and charged synthetic macromolecule.”
Unfortunately, there is no additional description concerning the twisting interaction in the revised version.
In the explanation of “3.2. Molecular Dynamic Simulation”, the authors explained the interactions adapted in the present study; 1) Excluded volume interaction (Lennard-Jones potential), 2) Bond connectivity between adjacent beads, 3) Harmonic bending potential, (bending potential), and Coulombic interactions (Debye-Huckel) Thus, it is quite obvious that authors completely ignored the effect of twisting interaction, which is generally very important for ring-shaped semiflexible polyelectrolytes, such as double-stranded DNA above the size of kbp. As the reviewer, it is quite strange why the authors added the statement of “The twisting interactions/potentials is introduced” in the revised version, against the actual model adapted for the simulation. Bending and twisting are physically different subjects each other as has been well explained in usual textbook of polymer physics.
Response: Thank you for pointing out the error. Indeed, there does not exist twisting potential in the model of ring PE. Bending potential and twisting potential are really/ different. Although there does not exist twisting potential in the model of ring PE, as the bending energy b increases, the twisting interaction enhances accordingly. In addition, the tetravalent cations which bind to ring PE may also cause localized twisting of ring PE, which can facilitate collapse. The twisting interaction enhances further.
With respect to the effect of counter ion condensation:
In the revised version, the authors newly added the following interpretation by adding references [39], [23].
The tetravalent cations can condense onto the ring PE backbone[39], while they drive away the monovalent counterions[23].
Unfortunately, this statement did never play the role as a reasonable response in the revision. If the authors carefully read and check the related article concerning the effect of “counter ion condensation” on the “condensation of polyelectrolyte”, they may recognize the very important and nonnegligible effect of the competition between monovalent and multivalent counter ions through the “counter ion condensation”. In other words, changes in the translational entropy of monovalent cations play the dominant role in the stability of the condensed polyelectrolytes. The reviewer would like to suggest the authors to learn the past representative studies concerning the effect of counter ions, as exemplified in the following.
- Schiessel, et al., Counterion-Condensation-Induced Collapse of Highly Charged Polyelectrolytes. Macromol., 31, 7953(1998). Cited by 275.
- A. Bloomfield, DNA Condensation by Multivalent Cations, Biopolymers, 44, 269(1997). Cited by 1399.
In the present study, the authors adapted the Debye-Huckel approximation. The validity of counter ion condensation hypothesis indicates the break-down of the framework of Debye-Huckel, owe to the higher-order correlation among ionic species. Thus, it is necessary for the author to take into consideration of the effect of the drastic change on the manner of counter ion condensation accompanied by the folding transition of polyelectrolyte chains.
Response: Thank you for your valuable comments. we have revised it in the revised manuscript.
Minor comments:
On Figures 1,3,9: For these pictures, the length scales are different each other. It may be better to show the scale bar with the unit of σ.
Response: We have added the scale bar in the revised manuscript.
On Figures 2A and 4A: Why are the horizontal and vertical scales different for each diagram?
Response: We take spindle with two racquet heads as example to explain it. The beads can not form contact in the regime of racquet head, while the beads form contact in the regime of spindle. we can observe that different beads form contact for the two different conformation of Figure 1 of attachment.
On Figures 5-8: The authors showed various plots, indicating that there are no bistability, or coexisting of different states, for the simulation with the same parameters. This seems unrealistic. Past experimental and theoretical studies indicated that different morphology coexists in many situations. For example, coexistence of toroid and rod is well known for the folded compact DNA, as exemplified in the following article.
- Golan, et al., DNA Toroids: Stages in Condensation, Biochem., 38, 14069(1999). Cited by 273
Response: The results of Golan show that the DNA molecules coil fold several times into progressively shorter rods which can open up into toroids. The results of Reddy show that a linear chain condenses into a toroid after the nucleation of a loop in the end of chain, while the chain fold into a spindle after the nucleation of a loop in the middle of chain[1]. The barriers for a fully condensed spindle to convert into a toroid is high. unfolding in poor solvent conditons is highly unfavorable[1]. Our results show that the ring PE draws close to form rod. The main reasons is that chain length is not long enough in our work. There coexist toroid and rod at the phase transition point between the rod and toroid.
- Dey, A.; Reddy, G. Toroidal condensates by semiflexible polymer chains: insights into nucleation, growth and packing defects. J. Phys. Chem. B. 2017, 121, 9291–9301. DOI:10.1021/acs.jpcb.7b07600

Round 3
Reviewer 1 Report
Comments and Suggestions for Authors
On the occasion of the second revision, the authors admitted the ignorance of the twisting interaction in their manuscript, by added the statement in the second revised version; Although there does not exist twisting potential in the model of ring PE, as the bending energy b increases, the twisting interaction enhances accordingly.
The reviewer would like to point out the essentially important effect of twisting for the circular semiflexible polymer chain. Nowadays, it has been well established that semiflexible polymer chain can never interpreted with the framework of Flory-Huggins model. One of the most important factors is the significant effect of twisting interaction. Most of the polymer physicists may agree that twisting interaction would have minor effect but should play major role for the determination of its conformation. As the reviewer has repeatedly indicated the importance of twisting for circular semiflexible chain, the topological characteristics in the condensed/compact conformation should be determined under the framework of ‘Linking number = Twist + Writhes’. Thus, the reviewer claims that the content of the present manuscript definitely missed the very important physical characteristic of twisting interaction in circular semiflexible polymer.
At least, it is the necessary for the authors to make clear the missing consideration of the twisting interaction by reconstituting the interpretations in the Title and Abstract.
Concerning the effect of counter ion condensation:
The authors newly added a sentence; The translational entropy of monovalent 249 counterions is produced, and it exerts an important effect on the folding of ring PE[40].
Unfortunately, there seems to be no explanation of the important effect for the conformation of the semiflexlble polymer chain adapted in the present study.
Additional minor comments:
The unit of Cs is missing in Figures 5 and 7.
It would be better for the authors to show the actual values for the persistence lengths, Bjerrum length, number of existing ions adapted in the present numerical simulations.
Comments on the Quality of English LanguageThere seems to be no serious problem in the linguistic quality.
Author Response
Comments 1:
On the occasion of the second revision, the authors admitted the ignorance of the twisting interaction in their manuscript, by added the statement in the second revised version; Although there does not exist twisting potential in the model of ring PE, as the bending energy b increases, the twisting interaction enhances accordingly.
The reviewer would like to point out the essentially important effect of twisting for the circular semiflexible polymer chain. Nowadays, it has been well established that semiflexible polymer chain can never interpreted with the framework of Flory-Huggins model. One of the most important factors is the significant effect of twisting interaction. Most of the polymer physicists may agree that twisting interaction would have minor effect but should play major role for the determination of its conformation. As the reviewer has repeatedly indicated the importance of twisting for circular semiflexible chain, the topological characteristics in the condensed/compact conformation should be determined under the framework of ‘Linking number = Twist + Writhes’. Thus, the reviewer claims that the content of the present manuscript definitely missed the very important physical characteristic of twisting interaction in circular semiflexible polymer.
At least, it is the necessary for the authors to make clear the missing consideration of the twisting interaction by reconstituting the interpretations in the Title and Abstract.
Response 1: We have added the effect of twisting interaction on forming toroid in the Abstract of revised manuscript. In fact, the twisting interaction changes with the bending energy in our study. If the twisting potential is introduced, the rigidity of PE is controlled by the bending potential and twisting potential. Therefore, it is very difficult to linearly adjust the rigidity of PE.
Comments 2:
Concerning the effect of counter ion condensation:
The authors newly added a sentence; The translational entropy of monovalent 249 counterions is produced, and it exerts an important effect on the folding of ring PE[40].
Unfortunately, there seems to be no explanation of the important effect for the conformation of the semiflexlble polymer chain adapted in the present study.
Response 2: we have provided explanation in the revised manuscript.
Additional minor comments:
Comments 3:
The unit of Cs is missing in Figures 5 and 7.
Response 3: Thank you. We have revised them in the revised manuscript.
Comments 4:
It would be better for the authors to show the actual values for the persistence lengths, Bjerrum length, number of existing ions adapted in the present numerical simulations.
Response 4: we have added them in the revised manscript. The tetravalent cations play decisive role in the folding process of ring PE. Each salt molecules contains one tetravalent cation and four anions. Therefore, the number of tetravalent cation is equal to the number of salt molecules. That is why we only provide the relation between the number of salt molecules and salt concentration in revised manuscript.

Round 4
Reviewer 1 Report
Comments and Suggestions for Authors
In response to the missing interpretation concerning the effect of twisting interaction (Comment1):
In the report 3 (the last comments), the reviewer claimed, “ At least, it is the necessary for the authors to make clear the missing consideration of the twisting interaction by reconstituting the interpretations in the Title and Abstract.” As the response, in the Abstract the authors added a sentence, “In addition, the twisting interaction plays an important role in ring PE’s forming toroid.” As for the similar modification, the authors newly added the following sentence on line 309; “The twisting interaction stabilize toroidal structure.” It is also found that authors added the interpretation on line 418, “the twisting interaction induce it to condense into toroid”.
These newly added sentences provide the impression for the readers that the authors performed the simulation including the twisting rigidity, which is completely against the actual content of the present manuscript. The reviewer can never suggest the acceptance of the present revised version for the publication; such intentional paraphrase given by the authors is against the scientific sincerity.
In the sentences of Response 1, the authors made an excuse, “If the twisting potential is introduced, the rigidity of PE is controlled by the bending potential and twisting potential. Therefore, it is very difficult to linearly adjust the rigidity of PE”.
The reviewer would like to suggest the authors to learn the past established concept both from theory and experiments that the conformation of the circular semiflexible polymers, including circular DNAs, should be interpreted based on both bending and twisting rigidities.
For example, the authors can learn from the past published articles, how twisting plays the essential effect to determine the conformation of semiflexible circular polymer chain:
Statistical mechanics of supercoiled DNA. J. F. Marko and E. D. Siggia, Phys. Rev. E, 52(1995). 342 citations
Entropic Elasticity of Twist-Storing Polymers. J. D. Moroz and P. Nelson, 31(1998). 173 citations
Structure of Plectonemically Supercoiled DNA. T. C. Boles, et al., J. Mol Biol., 213(1990). 334 citations
The Twist, Writhe and Overall Shape of Supercoiled DNA Change During Counterion-induced Transition from a Loosely to a Tightly Interwound Superhelix. J. Bednar, et al., J. M0l. Biol., 235(1994). 228 citations
The above publications indicate that the importance of twisting for the conformation of semiflexible polymer has been scientifically established more than two decades ago.
If the reviewer provides further suggestion on this issue, at least scientifically accurate representation should be given in Title and Abstract, as the reviewer claimed in the comment of round 3.
For example, the title should be renewed.
Conformational Transition of Semiflexible Ring Polyelectrolyte in Tetravalent Salt Solutions: A Simple Numerical Modeling without the Effect of Twisting
It may be also possible to add the interpretation in Abstract as exemplified as follows:
In the present study, we have performed molecular dynamic simulations for a beaded chain model. For simplification, here we have neglected the effect of twisting interaction, although it has been well known that both bending and twisting interactions play the deterministic for the steric conformation of a semiflexible ring polymer.
The reviewer regards that the revised manuscript has not been modified in a satisfactory manner even for the other issues, for example for the comments 2 and 4 in the last circle (Report 3). In this occasion, the reviewer claims the very important issue which should be improved before the final consideration of acceptance/rejection.
Comments on the Quality of English LanguageThe quality of the English language is not so bad. Minor revisions with respect to the linguistic may improve the manuscript.
Author Response
In response to the missing interpretation concerning the effect of twisting interaction (Comment1):
In the report 3 (the last comments), the reviewer claimed, “ At least, it is the necessary for the authors to make clear the missing consideration of the twisting interaction by reconstituting the interpretations in the Title and Abstract.” As the response, in the Abstract the authors added a sentence, “In addition, the twisting interaction plays an important role in ring PE’s forming toroid.” As for the similar modification, the authors newly added the following sentence on line 309; “The twisting interaction stabilize toroidal structure.” It is also found that authors added the interpretation on line 418, “the twisting interaction induce it to condense into toroid”.
These newly added sentences provide the impression for the readers that the authors performed the simulation including the twisting rigidity, which is completely against the actual content of the present manuscript. The reviewer can never suggest the acceptance of the present revised version for the publication; such intentional paraphrase given by the authors is against the scientific sincerity.
In the sentences of Response 1, the authors made an excuse, “If the twisting potential is introduced, the rigidity of PE is controlled by the bending potential and twisting potential. Therefore, it is very difficult to linearly adjust the rigidity of PE”.
The reviewer would like to suggest the authors to learn the past established concept both from theory and experiments that the conformation of the circular semiflexible polymers, including circular DNAs, should be interpreted based on both bending and twisting rigidities.
For example, the authors can learn from the past published articles, how twisting plays the essential effect to determine the conformation of semiflexible circular polymer chain:
Statistical mechanics of supercoiled DNA. J. F. Marko and E. D. Siggia, Phys. Rev. E, 52(1995). 342 citations
Entropic Elasticity of Twist-Storing Polymers. J. D. Moroz and P. Nelson, 31(1998). 173 citations
Structure of Plectonemically Supercoiled DNA. T. C. Boles, et al., J. Mol Biol., 213(1990). 334 citations
The Twist, Writhe and Overall Shape of Supercoiled DNA Change During Counterion-induced Transition from a Loosely to a Tightly Interwound Superhelix. J. Bednar, et al., J. M0l. Biol., 235(1994). 228 citations
The above publications indicate that the importance of twisting for the conformation of semiflexible polymer has been scientifically established more than two decades ago.
If the reviewer provides further suggestion on this issue, at least scientifically accurate representation should be given in Title and Abstract, as the reviewer claimed in the comment of round 3.
For example, the title should be renewed.
Conformational Transition of Semiflexible Ring Polyelectrolyte in Tetravalent Salt Solutions: A Simple Numerical Modeling without the Effect of Twisting
It may be also possible to add the interpretation in Abstract as exemplified as follows:
In the present study, we have performed molecular dynamic simulations for a beaded chain model. For simplification, here we have neglected the effect of twisting interaction, although it has been well known that both bending and twisting interactions play the deterministic for the steric conformation of a semiflexible ring polymer.
Response 1: Thank you for your help and good advice. The bending energy is regarded as a penalty for successive bonds deviating from a straight arrangement. Therefore, in our model, the bending interaction contains twisting interaction to some extent, the twisting interaction is indeed very important for the steric conformation of a semiflexible ring polymer. Therefore, we have revised the title and the abstract in the revised manuscript, according to your advices.
The reviewer regards that the revised manuscript has not been modified in a satisfactory manner even for the other issues, for example for the comments 2 and 4 in the last circle (Report 3). In this occasion, the reviewer claims the very important issue which should be improved before the final consideration of acceptance/rejection.
Response 2: we are very sorry for that. We revised them again in the revised manuscript.

Round 5
Reviewer 1 Report
Comments and Suggestions for Authors
The reviewer found that the authors have added important notations both in the Title and Abstract. Thus, the revised manuscript may become to exhibits the aspect toward the positive consideration for the publicatio.
Comments on the Quality of English LanguageThe linguistic quality seems to be not so bad. Careful check on the English presentation will improve the relaiability of the manuscript.